# Intracellular Routing and Recognition of Lipid-Based mRNA Nanoparticles

**DOI:** 10.3390/pharmaceutics13070945

**Published:** 2021-06-24

**Authors:** Christophe Delehedde, Luc Even, Patrick Midoux, Chantal Pichon, Federico Perche

**Affiliations:** 1Innovative Therapies & Nanomedicine, Centre de Biophysique Moléculaire CNRS UPR4301, Rue Charles Sadron, 45071 Orléans, France; christophe.delehedde@sanofi.com (C.D.); patrick.midoux@cnrs.fr (P.M.); 2Sanofi R&D, Integrated Drug Discovery, 91385 Chilly-Mazarin, France; luc.even@sanofi.com

**Keywords:** mRNA delivery, intracellular routing, lipid-based nanoparticles

## Abstract

Messenger RNA (mRNA) is being extensively used in gene therapy and vaccination due to its safety over DNA, in the following ways: its lack of integration risk, cytoplasmic expression, and transient expression compatible with fine regulations. However, clinical applications of mRNA are limited by its fast degradation by nucleases, and the activation of detrimental immune responses. Advances in mRNA applications, with the recent approval of COVID-19 vaccines, were fueled by optimization of the mRNA sequence and the development of mRNA delivery systems. Although delivery systems and mRNA sequence optimization have been abundantly reviewed, understanding of the intracellular processing of mRNA is mandatory to improve its applications. We will focus on lipid nanoparticles (LNPs) as they are the most advanced nanocarriers for the delivery of mRNA. Here, we will review how mRNA therapeutic potency can be affected by its interactions with cellular proteins and intracellular distribution.

## 1. Introduction

The in vivo administration of mRNA for in situ protein production was first reported in 1990, after intramuscular injection of an mRNA coding a reporter gene [1]. Since, there has been a growing interest in mRNA as a therapeutic modality, especially in the last decade [2,3]. It can be used as a platform for protein replacement therapies, genome engineering, cellular reprogramming, tolerization for allergy, and immunotherapies including vaccination [3,4]. Compared to plasmid DNA (pDNA), mRNA does not need to reach the nucleus, as it is expressed in the cytosol where the translation machinery resides. Hence, the mRNA approach is also safer, with no possibility of mutation, integration, or other undesirable genetic events [3]. In addition, it offers the possibility to transfect difficult-to-transfect differentiated cells, such as dendritic cells and neurons [5,6,7,8,9]. Moreover, mRNA is produced in vitro in a cell-free system, and methods and facilities for the large-scale production of therapeutic mRNA have been built to produce the approved SARS-CoV2 mRNA vaccines [10,11].

Since RNA molecules are unstable, prone to extracellular nucleases degradation, and are membrane-impermeable, mRNA delivery systems are required to protect them from degradation and to promote their cellular uptake into the targeted cells. Therefore, delivery systems allow them to be protected from rapid degradation in biological fluids as well as to enhance their cell uptake. In fact, it has been shown that free circulating RNA can be degraded within 15 s in the presence of blood RNAses (which are even more present in cancer patients), and mRNA has an estimated half-life of 1–2 min in human serum or bovine vitreous [12,13].

To date, different formulations have been designed to improve the stability and the intracellular delivery. Lipid-based systems represent the majority of those formulations, besides those comprising polymer, micellar or lipid–polymer hybrid [2,3,14,15,16]. In all cases, as their size ranges from a hundred nanometers to a few micrometers, these formulations are categorized as nanoparticles (NP). To deliver their cargo intracellularly, NP need to diffuse across the dense extracellular matrix and across the plasma membrane [17,18]. The main mechanism for NP to cross these barriers is endocytosis, where invaginations of the cell membrane confine NP in endosomal vesicles targeted for degradation [19]. Accordingly, NP need to incorporate features to control cell targeting, intracellular trafficking and the intracellular release of the mRNA in the cytoplasm for its translation [20]. Another important limitation to the delivery of synthetic mRNA is its recognition as foreign mRNA by intracellular sensors, such as Toll-like receptors (TLRs) 7 and 8. This recognition induces a type I interferon (INF α/β) response, which induces the suppression of mRNA translation as well as a detrimental inflammatory response [11,21]. Interestingly, in addition to mRNA, the chemical groups of the vector have been reported to be sensed by the immune system as well, as reported by Anderson’s lab [22].

The most advanced lipid-based NP for mRNA delivery are lipid nanoparticles (LNPs) (reviewed in [11,15,23]). LNPs are prepared by the microfluidic mixing of lipids in ethanol and mRNA in acidic buffer (pH ≤ 4.0). Lipids include an ionizable lipid (pKa < 7) that will be protonated at acidic pH to condense mRNA and release mRNA inside the cells, cholesterol for stabilization, and helper lipids for endosomal escape DOPE (1,2-dioleoyl-sn-glycero-3-phosphatidyl-ethanolamine) or DSPC (distearoylphosphatidylcholine) and a PEGylated (polyethylene glycol) lipid to prevent aggregation of LNPs [3]. LNPs combine the following several advantages: a high mRNA encapsulation efficiency, can be injected by various routes, preclinical and clinical proof of activity, and stability under storage (reviewed in [3,11,24]). Moreover, their production is reproducible and several techniques exist for the large-scale production of mRNA LNPs [25]. Accordingly, our review will focus on LNPs-mediated delivery of mRNA.

Despite the lack of correlation between in cellulo mRNA transfection and in vivo delivery efficacy [26,27], the evaluation of mRNA delivery systems in cell culture still represents the first step of the formulation design. Accordingly, understanding the correlation between intracellular bottlenecks and therapeutic activity is critical to further advance mRNA therapy [3,28]. There are very few studies that address the cellular uptake and subcellular distribution of exogenous mRNA [29].

The first step in mRNA delivery is to reach the targeted organ where the therapeutic mRNA should be expressed. Several units in the LNPs composition stabilize the LNPs to maximize the dose reaching the target cell. Such units should prevent sedimentation of LNPs in the storage medium, prevent aggregation in the physiological medium they are dispersed into after injection, limit drug release after interaction with the factors present in physiological fluids, and improve accumulation in the targeted organs and targeted cells. To grasp the complexity of the NP journey, we refer the reader to a review by AT Florence, describing the challenges faced by NP after administration into the body [30]. Decreasing mRNA leakage from LNPs and improving their circulation time have been the focus of extensive research (reviewed in [25]).

Indeed, after intravenous injection, most NP are captured by innate defense systems against exogenous agents [31]. Surface modification with lipids conjugated to hydrophilic PEG (polyethylene glycol) groups decreases protein adsorption onto LNPs and prevents their aggregation in the circulation [15]. As opsonization by serum-borne proteins marks NPs for elimination by cells of the mononuclear phagocyte system in the spleen and liver, PEGylation increases the circulation time of LNPs and their chance to reach the target organ [25]. The benefit of PEGylation is not limited to intravenous injection, as the decreased opsonization and neutralization of surface charge by PEG groups also favors the delivery to draining lymph nodes after intramuscular injection, which is a critical parameter for vaccination [32,33]. However, PEGylation also decreases the cellular uptake of NP and their endosomal escape ability [31]. The compromise between enough PEG-lipid to decrease non-specific interactions, yet not too much to allow endocytosis, has been referred to as the “PEG dilemma” [34]. Further, mRNA LNPs usually contain 1.5% PEGylated lipid, which is a percentage that is sufficient to avoid LNPs aggregation during storage and limit protein adsorption after infusion [11,15,23,24,35]. To resolve the PEG dilemma, ligands are also attached to a fraction of PEGylated lipids for cellular targeting. Stimuli-sensitive PEGylation, with PEGylated units activated by stimuli enriched in the target organ, are another solution to the PEG dilemma (reviewed in [36]).

The other critical parameter for the extracellular stability of LNPs is the presence of cholesterol, which is a feature already reported essential to decrease drug leakage from liposomes [37]. Cholesterol increases the viscosity of the surface and increases the encapsulation efficiency (reviewed in [25]). The presence of cholesterol impedes lipid extraction by high-density lipoproteins, which would cause LNPs breakdown and mRNA leakage. Finally, the ionizable lipid ensures the complexation of mRNA in a segregated region of LNPs, protecting it from degradation by extracellular nucleases [35,38]. In this review, we describe different approaches to study cellular uptake, the endosomal escape, and intracellular sensing of delivered mRNA (Figure 1).

## 2. Intracellular Trafficking of mRNA

Although passive diffusion is suitable for small drug molecules, mRNA needs a vector for efficient uptake by the cell. As for plasmid DNA, the uptake and intracellular trafficking of mRNA depend on the type of NP. Once bound to the surface, NP can be internalized via multiple pathways. The two main endocytic pathways are phagocytosis and pinocytosis (Figure 2) [39]. Phagocytosis is not the most studied endocytic pathway for NP, since it is limited to immune cell types [40]. It is mainly described as a mechanism for large molecules, with up to micrometer range. When a particle is engulfed by phagocytosis, it is entrapped into a phagosome that will be fused to a lysosome to form a phagolysosome. Some receptors, such as mannose, fructose and scavenger receptors, can play a role in phagocytosis [41]. In comparison to phagocytosis, pinocytosis is present in the majority of cells, and it comprises clathrin- and caveolin-mediated endocytosis, which are two of the most described endocytic pathways for NP.

Here, we focus on the main pathways described for mRNA NP, which are clathrin- and caveolae-mediated endocytosis, macropinocytosis, and phagocytosis [42].

The clathrin-dependent route is energy dependent and involves a clathrin network. Upon the binding of NP at the cell surface, there is a membrane invagination and the formation of clathrin-coated vesicles that will fuse with early endosomes after elimination of the clathrin network [42]. From early endosomes, NP can be sorted either to the endoplasmic reticulum of the Golgi network for retrograde transport, to the recycling endosome for exocytosis, or to the late endosome/lysosome for degradation [43]. Most of the internalized molecules are then transferred to the late endosomes and then to the lysosomes. A pH gradient is established within these vesicles, ranging from the less acidic, early endosomes (pH 6.5–6.8) to the most acidic, lysosomes (pH 4.5) [44]. The vesicular trafficking is controlled by Rab proteins, which are small GTPases that are specific for each type of vesicle, as Rab5 and Rab7 for the early and late endosomes, respectively.

The caveolae-mediated pathway appears to be limited to NP smaller than 100 nm [45]. In this pathway, instead of a clathrin network, the invaginated vesicle, termed caveola, possesses a network of caveolin 1. Caveolae are present in particular in lipid islets (lipid rafts) rich in cholesterol and sphingolipids, mainly found on endothelial cells. This route is non-acidic and non-digestive. Indeed, after engulfment of cargo, caveosomes were reported to fuse with lysosomes later than clathrin-coated vesicles, thereby favoring transfection with nucleic acids [46,47,48].

Macropinocytosis allows the internalization of fluids by the extension of a plasma membrane that is supported by actin filaments and regulated by Rho GTPases. In mammals, phagocytosis is particularly developed in macrophages, monocytes and neutrophils, to eliminate bacteria, yeasts, apoptotic bodies and fatty deposits. This is an active process involving specific cell receptors and intracellular signaling cascades mediated by Rho GTPases.

Plasmalemma vesicle-associated protein 1 (PV1) is a caveolae-associated protein expressed by the endothelial cells of the lungs [49]. A recent report optimized endothelial cells targeted by the conjugation of an anti-PV1 antibody to MC3-based mRNA LNPs [50]. Using 70 nm LNPs conjugated with a PV1 antibody or an isotype control, they reported a 14-fold increase in lung accumulation of targeted LNPs, mirrored with a 24-fold increase in mRNA expression in the lungs over untargeted LNPs. However, no functional mRNA targeting was observed, whereas PV1-targeted LNPs predominantly accumulated in the lungs (lung-to-liver ratio of 2:1); a two-fold higher luciferase expression in the liver over the lungs was detected, indicating a difference in the biodistribution and expression of targeted LNPs. They used larger LNPs of 160 nm to overcome this lack of selective mRNA expression in the lungs. Their assumption was based on the fact that the flexibility of the LNPs would be able to overcome the 100 nm size limit of PV1-mediated endocytosis [51]. With conjugation of the anti-PV1 antibody to 160 nm, the LNPs increased their lung accumulation, and this resulted in a 50-fold increase in luciferase expression in the lungs over untargeted 160 nm LNPs. Interestingly, 160 nm targeted LNPs showed a lung-to-liver accumulation ratio of 2.5- and 40-fold more expression of mRNA in the lungs over the liver. These data demonstrate that functional mRNA targeting the lungs requires selection of the ligand and of the NP size.

Note that most complexes between chemical vectors and nucleic acids are positively charged, and, depending on the type of vector, they would be internalized by clathrin-dependent endocytosis or by caveolae after electrostatic interactions with proteoglycans and glycosaminoglycans, which are anionic macromolecules present on the external face of the plasma membrane [52,53].

Among the different techniques available to study the cellular uptake and trafficking of nanoparticles, flow cytometry appears fast and informative [54]. Using fluorescein-labeled NP, different information can be obtained regarding particle localization on the cell surface, inside the cell or into acidic compartment, in which the acidic pH quenches the fluorescence of fluorescein [55]. For that, cells must be transfected with mRNA/vector complexes bearing fluorescein (i.e., fluorescein lipid or fluorescein-labeled mRNA). After incubation, the cells are harvested without trypsin to avoid the detachment of the complexes present on the surface. Then, three measures of cell-associated fluorescence are recorded to consider the following: (1) total cell fluorescence (surface and intracellular), (2) intracellular fluorescence after trypan blue (TB) treatment and, (3) fluorescence corresponding to NP in acidic compartments after monensin treatment (Figure 3). In Figure 3, we present typical results obtained with lipopolyplexes—nanoparticles made of hybrid polymer and liposome mRNA complexes—incubated with dendritic cells. The total cell-associated fluorescence intensity (Figure 3A) corresponds to the sum of the fluorescence of internalized NP, and cell surface-associated NP quantified following TB treatment (Figure 3B) [56]. TB-mediated elimination of cell surface fluorescence is due to the fact that TB is expelled by viable cells and it is known to quench the green emission of some fluorophores, among which are FITC (fluorescein isothiocyanate) and NBD-PE (1,2-dipalmitoyl-sn-glycero-3-phosphoethanolamine-N-(7-nitro-2-1,3-benzoxadiazol-4-yl)), which are frequently used to label mRNA and lipids, respectively [57]. To evaluate the relative distribution of NP inside acidic compartments (Figure 3E), a reliable method is to compare the cell-associated fluorescence intensity before and after monensin ionophore treatment [54]. This method takes advantage of the environmental pH-dependent fluorescence of fluorescein, with a lower intensity at acidic pH, [58] and of the ability of monensin to neutralize the pH of acidic compartments such as endosomes and lysosomes [54]. An increase in fluorescence (usually 1.5–3-fold) in monensin-treated cells indicates a change in the environment from acidic pH to neutral pH, mirroring the presence of NP in acidic organelles.

This technique was used to compare the trafficking patterns of cationic mRNA–lipopolyplexes (LPR) bearing, or not, a tri-mannose (TM) lipid in dendritic cells, which express the mannose receptor [59]. Cationic LPR made with a TM-lipid [60] exhibit lower cell surface-associated fluorescence than LPR devoid of targeting moiety (Figure 3C), suggesting a lack of specificity attributed to the non-specific interaction of the positively charged LPR with the negatively charged cell membrane. Oppositely, intracellular fluorescence is more than two times higher in the presence of TM (Figure 3D), indicating that the presence of TM in LPR enhances intracellular uptake in dendritic cells. Furthermore, Figure 3E shows a higher monensin increase for TM complexes, corresponding to its presence in more acidic compartments. In fact, it is known that mannose receptors induce clathrin-mediated endocytosis [61] going faster in an acidic endosome in comparison to caveolin-mediated endocytosis, which seems to be the case for non-TM LPR with non-specific uptake.

Another cellular component that must be considered for NP uptake is sulfated proteoglycans. They are among the most negatively charged macromolecules that affect cell uptake. They consist of a protein core covalently linked to the following glycosaminoglycans: heparin, heparan sulfate, dermatan sulfate, chondroitin sulfate, and keratan sulfate [17]. The burden of proteoglycans is mainly due to the sulfated groups, and this sulfation is inhibited by sodium chlorate. The participation of these molecules during in cellulo and in vivo transfection by chemical vectors has been demonstrated following treatment with heparinase I, or the preincubation of cells with negatively charged polysaccharides [62,63].

It is accepted that clathrin-mediated uptake is faster than caveolin; thus, targeting the caveolin pathway should lead to a more productive delivery, with more time to escape endosomes than clathrin-mediated uptake, which can lead to major accumulation in late endosomes and lysosomes. The endocytosis pathway can also depend on factors such as the chemicals structures of the NP, or the presence of ligands; therefore, it can also be affected with the cell types. For instance, HepG2 cells lack endogenous caveolin [64,65]. Alternatively, intermediately mature dendritic cells can contain non-lysosomal, mildly acidic, class II vesicles, which are Rab7- (Ras-related protein, associated with the late endosome), but Lamp-1+ (lysosome-associated membrane protein 1) [66].

### 2.1. Endosomal Escape of Nanoparticles

When endocytosis occurs, the cell starts to engulf a part of its plasma membrane to entrap the NP inside small vesicles (coated with clathrin, caveolin, etc.) in combination with extracellular fluids. Once invaginated, those vesicles are pulled together after a tethering mechanism into an organelle, which is the early endosome [41,67]. During their journey in the endosomes, the NPs are subjected to a pH gradient starting from neutral extracellular pH (7.4), followed by gradual acidification in early endosomes (pH 6.3), late endosomes (pH 5.5) and finally lysosomes (pH < 5) [68]. To avoid degradation in the lysosomes, mRNA nanocarriers need to escape from the endosomes to reach the cytosol (pH 7.2) for mRNA translation [3]. The strategies used to escape this terminal degradation rely on units activated by acidic pH [20,69]. Further, mRNA nanocarriers incorporate ionizable units and/or fusogenic lipids to destabilize the endosomal membrane, and hence deliver mRNA into the cytosol [3].

After cellular uptake, mRNA needs to escape the endosomes, which is a limiting step for productive mRNA delivery, as only ≈ 1–2.5% of mRNA was detected in the cytosol after transfection of human epithelial cells with mRNA LNPs made of the ionizable lipid D-Lin-MC3-DMA [70,71]. This drastic limiting step led to the development of formulations with different strategies for enhanced intracellular delivery and endosomal escape (Table 1).

Optimization of the ionizable lipid tail is a critical parameter for the endosomal escape of mRNA delivery systems [3,26]. A systematic examination of a library of ionizable lipids harboring the same polar head and saturated alkyl acrylate chains, from 6 to 18 carbon length, identified 10 carbons as the best length [26]. A pattern was confirmed with two different polar heads. Although LNPs with 6 to 18 carbons exhibited similar biodistribution after intravenous injection of cyanine 5-labeled luciferase mRNA, only LNPs formed with a tail of 10 carbons led to detectable luciferase expression in mice liver and spleen. The in vivo efficacy was dictated by the influence of carbon length on the amplitude of protonation, not by LNPs size or mRNA entrapment efficiency. By measuring protonation of the different LNPs, at a pH corresponding to endosomes (pH 5), a strong relationship between protonation and in vivo activity was evidenced, suggesting a critical role of the lipid tail during LNPs formation and internal organization of LNPs upon endosomal escape.

A complementary study evaluated the saturated or unsaturated lipid tails of six or eight carbons for LNP-based mRNA delivery [74]. They prepared ionizable lipids with the 4A3 amine core and 13 different lipid tails. LNPs were formulated using a mix of ionizable lipid, DOPE, cholesterol, and DMG-PEG (dimyristoyl glycerol-PEG) (15:15:30:3, molar ratios). They reported minor differences for the in cellulo transfection of IGROV-1 human ovarian cancer cells of lipid tails of six or eight carbons, either saturated or unsaturated (one or two unsaturations). However, no transfection was detected when the ionizable lipid contained a farnesyl tail (11 carbons and 3 unsaturations), suggesting a role for carbon length and rigidity in the resulting transfection. After intravenous injection of LNPs prepared with luciferase mRNA in mice (0.25 mg/kg), series with eight carbon tails showed higher mRNA expression over series with six carbons tails. Amongst the eight carbons series, a similar expression was obtained with LNPs containing unsaturated tails (SC8) or one unsaturation (citronellol tail). However, lipids with two unsaturations (8/2 or nerol) exhibited lower luciferase expression, suggesting the chemical structure of the lipid rather than the tail length as a driver for efficacy. Analysis intracellular distribution by confocal microscopy revealed a three-fold lower colocalization of SC8 or citronellol LNPs over farnesyl ones, confirming the dominant bottleneck of endosomal escape for in vivo functional mRNA delivery. Altogether, these results highlight the requirement for a fine balance of lipid tail unsaturation; unsaturation is required to impart endosomal escape, but the structure should remain flexible, that is, not possess more than two unsaturations, to avoid rigidity.

An interesting study shows the importance of the endosomal escape for in vivo mRNA transfer, after eye delivery [75]. Patel and Ryal et al. demonstrate that the use of a different cationic or ionizable lipid (CIL) into the LNPs formulation can play a role in endosomal escape in vivo, after eye delivery of LNPs-mRNA encoding for reporter gene. A screening of LNPs prepared with different polar heads highlighted the different potentials for retinal pigment epithelial cells transfection [75]. Several polar heads comprising D-Lin-MC3-DMA (MC3), D-Lin-KC2-DMA (KC2), DODMA (1,2-dioleyloxy-3-dimethylaminopropane), DOTMA (1,2-di-O-octadecenyl-3-trimethylammonium propane), and DOTAP (1,2-dioleoyl-3-trimethylammonium-propane) were compared. This selection covers the most frequent cationic lipids used in LNPs for mRNA delivery, as it includes unsaturated cationic ionizable lipids with tertiary amines (group I: MC3, KC2, DODMA) and unsaturated cationic lipids with quaternary amines (group II: DOTMA, DOTAP). Although, the intraocular delivery of group I and II LNPs yielded detectable radiance at the organ level, and only group I LNPs were able to transfect retinal pigment epithelial cells. The lack of transfection using group II LNPs was due to their lower endosomal escape efficiency, and likely to their higher dissociation in biological fluids due to their cationic charge opposed to the neutral charge of group I LNPs.

Another study screened the polar heads of lipidoids for in vivo T lymphocytes transfection. [76]. They used different imidazole analogues and lipid tails to form LNPs with a mix of lipidoid, cholesterol, DOPE and DSPE-PEG (16:4:1:1, *w*/*w*). The first library of polar heads established that lipidoids containing imidazole were good for luciferase mRNA delivery to primary human T lymphocytes. To further optimize T cell transfection, they screened imidazole analogues and different unsaturated lipid tails. The lipids tails had different lengths (12 to 18 carbons), different linkers (acrylate or acrylamide) and carbon or heteroatoms in the tail (C, O, S, Se or SS disulfide). The best transfection of primary T cells was achieved with 17 carbons tails containing heteroatoms (O, S or Se). A second step consisted of screening imidazole analogues with different lengths, structures (branched or straight), or imidazole ring analogues. No CD8^+^ T cell transfection was obtained with imidazole ring analogues, confirming it as a key structure for mRNA delivery in T cells. Amongst imidazole analogues, an improvement in mRNA delivery was observed for the branched spacers of three carbons, with a ≈1.5-fold luciferase expression over the imidazole polar head. The identification of the best imidazole analogues was followed by a more detailed screening of lipid tails, with a comparison of the first atom in the tail (O or N), tail length (16–18 carbons), and the presence of heteroatoms (C, O, S, SS). LNPs with oxygen as the first atom in the tail worked more efficiently than those with nitrogen. In addition, the best tails contained 17 carbons and heteroatoms (O or S). The transfer of EGFP mRNA was performed to quantify the percentage of primary T cells transfected rather than just measuring luminescence in bulk. They compared LNPs formed with two kinds of heads, either imidazole (93), or imidazole with branching at the 2-imidazole position (9322) and with 17 carbons tails, with oxygen as the first atom and S heteroatom (39-O17S and 9322-O17S). The transfection efficiency reached 7% with 93-O17S, and 11% with 9322-O17S. Note that that the transfection capacities of the imidazole analogues, and of the different lipid tail structures, could not be predicted by in vitro screening, as they all had similar pKa and their membrane disruption capacity was not predictive of transfection efficiency. When 93-O17S and 9322-O17S were injected intravenously (0.6 mg/kg of luciferase mRNA), they resulted in detectable mRNA expression in the spleen. Finally, they used a transgenic mice model to decipher the in vivo activity of their LNPs. Cre recombinase mRNA was delivered in the Ai14 mice model that has a genetically integrated stop codon flanked by two *loxP* sites, upstream of the red fluorescent protein tdTomato gene [80]. In this model, successful Cre mRNA delivery will excise the DNA between the *loxP* sites, inducing tdTomato fluorescence. The combination of Cre mRNA delivery with immunostaining allows the identification of mRNA transfected cells. Further, 93-O17S and 9322-O17S LNPs were able to transfect all the following major immune splenocytes: 8% CD4^+^ T cells, 6% CD8^+^ T cells, ≈ 4% B cells, and 8% macrophages or dendritic cells. Such effective mRNA delivery in distinct immune cell types after intravenous injection opens up new pathways for mRNA immunotherapy.

Analysis of endosomal escape is critical to differentiate cellular uptake of mRNA and its expression, a difference also observed after intravenous injection of mRNA LNPs [81]. After the intravenous injection of the LNPs formed with the cKK-E12 ionizable lipid, Sago et al. reported lower accumulation of LNPs in the liver endothelial cells compared to hepatocytes and Kupffer cells [81]. Yet, flow cytometry analysis revealed that more endothelial cells expressed reporter mRNA over Kupffer cells and hepatocytes, highlighting a difference between biodistribution and functional expression in vivo, which was attributed to the differences in intracellular RNA trafficking among cell types.

Using a novel endosomal escape, a Renilla luciferase-based molecular probe, Saltzman’s group compared the correlation between mRNA expression levels and either cellular uptake or endosomal escape [82]. They used a library of 31 poly(amine-co-ester) (PACE) terpolymers containing a cationic diol for mRNA complexation and a lactone hydrophobic group to promote the formation of micelles, linked by biodegradable ester bonds. The PACE polymers in the library had different end groups and contained a panel of polymers with low, middle and high mRNA transfection efficiency in highly transfectable Expi293F human cells. The endosomal escape probe is a Renilla luc variant that is inactive and glycosylated [83]. The restoration of activity for this deglycosylation-dependent variant (ddRLuc) depends on the exclusively cytosolic enzyme N-glycanase-1 (NGLY1). The authors co-encapsulated fluorescently labeled firefly luciferase mRNA and ddRLuc to quantify and correlate cellular uptake (fluorescence), endosomal escape (Rluc luminescence), and mRNA expression levels (Fluc luminescence). The linear regression of mean fluorescence intensity with Fluc levels revealed a weak correlation (R² = 0.15), illustrating that uptake alone is not a good predictor of mRNA transfection efficiency. However, the linear regression of Rluc luminescence with Fluc levels demonstrated a high correlation (R² = 0.76), corroborating that endosomal escape success governs mRNA transfection efficiency. Developing new techniques or compounds to enhance the endosomal escape is then crucial to improve the efficacy of mRNA LNPs.

Disulfide bonds are frequently added in delivery systems to induce their intracellular dissociation, as the intracellular concentration of glutathione (GSH) is 1000-fold higher in the cytoplasm over the extracellular compartment [84]. The Harashima group designed a series of lipid-like materials named ssPalmO (ss-cleavable pH-activated lipid-like material), combining disulfide bonds (ss), oleic acid (O), and a tertiary amine for mRNA complexation and endosomal escape [72,73]. In these particles, the disulfide bonds are cleaved by intracellular GSH. However, in the first generation of ssPalm, only 25% of the disulfide bonds were cleaved. In the second generation of ssPalm, a self-degradable ionizable lipid (ssPalm-O-Phe) was included for increased cleavage of the disulfide bonds, based on “Hydrolysis accelerated by intra-Particle Enrichment of Reactant HyPER” [73]. LNPssPalm-O-Phe contain both nucleophiles (thiol groups) and electrophiles (phenyl esters). The thiols are only formed after GSH-mediated cleavage of disulfide bonds, and they lead to self-degradation by nucleophilic substitution. They prepared LNPs with degradable or non-degradable lipids and erythropoietin (EPO) mRNA. LNPssPalm-O-Phe yielded the highest EPO blood levels, even higher than the levels obtained with LNPs prepared with MC3 lipid. The improved in vivo expression with ssPalmO-Phe was attributed to the more complete and irreversible cleavage over ssPalmO-Ben or ssPalmO (Figure 4).

Finally, one should also consider that even with the same delivery system, modified and unmodified mRNA can have different intracellular distributions [29]. Using confocal microscopy, Kirschman et al. quantified free cytosolic mRNA, mRNA colocalized with stress granule makers or endosomal markers after the delivery of either unmodified or 5 mC/Ψ-modified mRNA, delivered with Lipofectamine 2000©. Although more free cytosolic mRNAs were detected 5 h post-transfection of 5 mC/Ψ-modified mRNA than unmodified mRNA, no difference was observed at later time points (12 h and 24 h). This suggests a faster endosomal release of modified mRNA and a higher entrapment of unmodified mRNA in stress granules, leading ultimately to lower expression of unmodified mRNA.

Moreover, if endosome escape is crucial, so is escaping kinetic, as it has been shown that after successfully escaping the endosome, mRNA faces a crowded environment once in the cytoplasm, which limits its free diffusion and is a source of protein–mRNA interactions that will affect its intracellular fate [85,86].

To establish correlations between intracellular trafficking kinetics and mRNA LNP transfection efficiency, Sayers et al. used cyanine 5-labeled mRNA encoding EGFP and cells with fluorescently labeled organelles [87]. They compared cell culture models with different susceptibilities to transfection; for example, highly permissive human colon epithelial cells (HCT116), permissive human lung epithelial cells (H358), and refractory murine colon fibroblasts (C26.WT). Correlating intracellular trafficking with EGFP mRNA expression (both intensity of EGFP and percentages of transfected cells), their study revealed a major contribution of endolysosome kinetics for successful mRNA transfection. Using a commercial dextran pH probe, they analyzed the pH exposure of internalized particles in endosomes. The CT26.WT cells presented the slowest kinetics of endosome acidification and the lowest difference in probe pH between 30 min and 24 h after transfection; H358 exhibited a fast acidification and the highest difference in probe pH between 30 min and 24 h; in HCT116 cells, the acidification rate was intermediate and reached a final pH between those found in CT26.WT and in H358 cells. These acidification kinetics were correlated with mRNA expression kinetics after transfection, as follows: a very slow increase in transfected cells percentage, from 0% at 4h to 5% at 36 h in CT26.WT; a slow increase in transfected cell percentages in H358 cells, from 2% at 4 h to 40% at 24 h and 50% at 36 h, and a continuous increase in transfected cell percentages in HCT116, from 50% at 4 h to 80% at 24 h and 90% at 36 h. Overall, their data suggest the requirement of an exposure of mRNA–LNPs to a sufficient decrease in pH, allowing protonation, and hence endosomal escape and transfection, but with kinetics permitting escape before fusion with lysosomes.

To decipher the contribution of endosomal compartments on mRNA–LNPs transfection in cellulo, Patel et al. used cell lines devoid of key endosomal trafficking proteins to decrease the biogenesis of either early endosomes (Rab5A), late endosomes (Rab7A), or recycling endosomes (Rab4A) [88]. Although the depletion of Rab5A or Rab4A had little influence on mRNA transfection, the depletion of Rab7A reduced mRNA expression by 30 to 70%, with the maximal inhibition of transfection at a high mRNA dosage (Figure 5). Electron microscopy imaging of cells depleted of Rab7A confirmed perturbation of late endosomes and lysosomes biogenesis with enlarged lysosomes (540 nm versus 90 nm in wild-type cells). Given the role of the lysosomal surface-associated protein mTORC1 (mechanistic target of rapamycin complex 1) on mRNA translation [89,90], they suspected decreased mTORC1 signaling as an explanation. In agreement with their intuition, the inhibition of mTORC1 by a pharmacological inhibitor decreased mRNA translation. On the contrary, constitutive activation of mTORC1 by knockdown of its regulator TSC2 (tuberous sclerosis complex 2) enhanced mRNA translation. Based on this finding, they next screened a library of 212 compounds with effects on lipids and vesicular trafficking, and identified MK571—a leukotriene inhibitor—as a booster of mRNA transfection. LNPs incorporating MK571 increased mRNA transfection by three-fold in HeLa cells and five-fold in HepG2 cells over LNPs. Note that the transfection efficiency of LNPs-MK571 was reduced in Rab7A-devoid cells, confirming a role of the lysosomal membrane in this booster effect. Moreover, after intravenous injection in mice, LNPs-MK571 allowed two-fold higher gene expression in the liver and spleen over LNPs. These data identify lysosomes as essential compartments for mRNA transfection and indicate that studies on intracellular mRNA trafficking should not be limited to early endosomal escape, but should be extended to the interaction with the lysosomal membrane.

Using single-molecule fluorescence in situ hybridization (smFISH) and co-staining with endosomal markers, Sabnis et al. measured the endosomal escape of several mRNA–LNPs by comparing cytosolic and total intracellular mRNA signals [71]. They reported an endosomal escape efficiency of 2.5% for MC3-LNPs. A screening of amino lipid series of ionizable lipids led to the identification of lipid 5, with a dramatic 15% endosomal escape in HeLa cells. This six-fold increase in endosomal escape was attributed to the lower Tm onset of lipid 5, a critical parameter for LNP fusogenicity [91]. Intravenous injection of lipid 5-LNPs, prepared with erythropoietin (EPO) mRNA in cynomolgus monkeys (0.1 mg mRNA/kg), resulted in five-fold higher plasma levels of EPO over MC3-LNPs, without the toxicity of MC3-LNPs, establishing the therapeutic potential of this ionizable lipid for mRNA delivery.

After endosomal escape, only a fraction of the internalized mRNA reaches the cytosol [3,70,87]. Indeed, a considerable proportion of the delivered mRNA is exocyted, which is a fate also reported for siRNA delivery [70,92]. Extracellular vesicles are a mode of endogenous RNA exchange between cells [93], and this mechanism of paracrine nucleic acids transfer was also described for transfected mRNA and siRNA [70,92]. A recent study reported that transfection of human epithelial HTB-177 cells with human erythropoietin (hEPO) mRNA LNPs led to the production of 120 nm extracellular vesicles containing mRNA–LNPs, designated mC3-EVs [70]. The mC3-EVs were able to transfect a culture of human PBMCs, which are difficult to transfect. Intravenous injection of mC3-EVs, from LNPs-transfected cells, led to the internalization and production of hEPO in the heart, lung, liver and spleen, with detectable hEPO levels in mice blood as soon as 2 h after injection. Interestingly, although hEPO levels were lower than direct injection of hEPO mRNA LNPs, mC3-EVs induced lower levels of inflammatory cytokines in the blood. Accordingly, mC3-EVs appear to be a safer process of mRNA delivery in multiple tissues.

The imaging of cells transfected with mRNA nanocarriers suggests that efficient translation is associated with mRNA localization close to the nucleus [87]. After endosomal escape, another critical parameter dictating translation is the intracellular mobility of mRNA [77]. A comparison of the intracellular fate of LNPs prepared with either cholesterol or the cholesterol analog β-sitosterol (βS) showed that both the LNPs had similar cellular uptake and endosomal escape. However, nanoparticle tracking analysis revealed a higher rate of endocytosis and an increased fraction of mobile βS LNPs over cholesterol ones, yielding a 48-fold enhancement in transfection efficiency over cholesterol-LNPs (Figure 6). This phenomenon was attributed to the different ultrastructural features of βS-LNPs, notably a high lamellarity, membrane rigidity, and being faceted, resulting in phase separation of lipid domains and enhanced fusogenicity [94].

### 2.2. Cell Sensors: Inflammatory Effect of mRNA NP

The vaccine has shown promising results in the past few years, thanks to the flexibility of the mRNA to encode for various antigens [95,96,97]. Recently, the context of COVID-19 has emerged a new interest regarding vaccination with mRNA–LNPs.

To induce potent antitumor responses with the vaccine, T cells must be activated against a specific antigen, through MHC presentation, which are present on the surface of antigen-presenting cells (APCs). In vivo, dendritic cells (DCs) are of importance, since they are the only APCs that can prime naïve T cells. Therefore, the aim of an mRNA vaccine will be to transfect DCs, leading to T cell activation [98]. However, mRNA transfection activates immune sensors, among which are the endosomal Toll-like receptors (TLRs) and the type I interferon pathway, leading to the production of inflammatory cytokines that can decrease mRNA translation via RNA-dependent protein kinase (PKR) phosphorylation of the translation initiation factor eIF2α, and promote mRNA degradation [3,99]. In fact, it has been documented that DNA and RNA can be sensed by Toll-like receptors (TLR 3, TLR7, TLR8, TLR9, TLR 13) [100] and RIG-I-like receptor (RLR), which can lead to an inhibition of the immune response and translation through the up-regulation of protein kinase R [14], as well as an excess of type I IFN that can induce proinflammatory cytokine, which can be deleterious [101].

If the inclusion of alternative nucleosides that are found in cellular mRNA, such as 5 mC or Ψ, and the removal of short RNA fragments by HPLC purification can resolve sensor activation [21,102], up-to-date in vitro synthesis of mRNA remains more expensive with modified nucleotides. Thus, approaches are developed to allow the use of unmodified mRNA without triggering the deleterious sensor effect. Ohto et al. compared the potentiation of unmodified mRNA LNPs transfection using the following two inhibitors: ISRIB (integrated stress response inhibitor), a specific inhibitor of eIF2α phosphorylation, and dexamethasone (DXM), a potent steroidal anti-inflammatory drug [78]. They treated mouse embryonic fibroblasts with either ISRIB or DXM, and transfected them with LNPs prepared with luciferase mRNA. Even though ISRIB and DXM did not influence cellular uptake and viability, they both increased mRNA translation with different profiles. ISRIB increased luciferase expression by 1.4-fold, 6 h after transfection, followed by a rapid return to radiance levels in the absence of ISRIB. Oppositely, DXM induced a long-lasting enhancement of luciferase expression from 4 h to 20 h. Next, they synthetized a DXM-palmitate conjugate to co-deliver DXM and mRNA. The DXM-palmitate was incorporated into LNPs without modifying their size, zeta potential, nor encapsulation efficiency. Intravenous injection of mRNA LNPs containing DXM, at a dosage of 0.25 mg/kg of mRNA and 0.65 mg/kg of DXM, improved luciferase expression in mice liver by 6.6-fold over mRNA LNPs. Altogether, this study represents an alternative to chemically modified nucleotides, a strategy that could be integrated in all lipid-based mRNA formulations.

Recently, a new interest has emerged regarding the lipid structure to act as an adjuvant of the immune response, which can lead to an enhanced immunity response. The adjuvant effect of the mRNA vaccine should be balanced, to give a strong innate immune response without activating the systemic activation of the immune system.

Zhang et al. developed LNPs nanovaccines based on a cationic lipid-like material that can efficiently deliver mRNA to APCs with a simultaneously adjuvant effect, through the activation of Toll-like receptor 4 (TLR-4) to induce T-cell activation. In comparison to heterocyclic lipids, the cationic lipid-like material is based on ring-open epoxides by generation 0 of poly(-amidoamine) (PAMAM) dendrimers [79]. They transfected mouse dendritic cells (DC2.4) with LNPs containing OVA mRNA, then coincubated the transfected DCs with B3Z cells (OVA-specific mouse CD8 T cell hybridoma), which secrete interleukine-2 (IL-2) upon OVA antigen stimulation. LNPs made of a C1 lipid-like material exhibit the highest IL-2 secretion, which results in efficient mRNA delivery into the DCs (IL-2 secretion was three-fold higher compared to Lipofectamine 2000, a commercially available transfection reagent). The same results were obtained with primary cells. C1-based LNPs (mRNA loaded or empty) show upregulation of the DCs surface and co-stimulatory molecules such as CD80, CD86, CD40, and an induction of the expression of pro-inflammatory cytokine genes such as IL-1b, IL-6, IL-12a and IL-12p40, indicating maturation and activation of DCs. When tested on BDMCs from wild-type and STING-KO mice, the expression of costimulatory molecules and type I IFN were similar, showing an induced immune activation independent of the STING pathway. The use of small inhibitors of TLRs (CU-CPT4a for TLR3; TAK-242 for TLR4; ODN2088 for TLR9; ODN2088 control for TLR7/8) exhibit that the TLR4 inhibitor blocked cytokine production, where other inhibitors partially blocked it, and the same result was observed in BMDCs KO for TLR4. An in vivo study was carried out to decipher antitumor efficacy. B6-mice were treated with either PBS, or OVA protein with aluminum salt (OVA + alum), or C1-OVA mRNA LNPs, and seven days later they were inoculated with MC38-OVA (OVA-expressing colorectal cancer cell line) or B16-OVA (melanoma cancer cell line). The results show that the best tumor growth inhibition was obtained with C1-OVA LNPs in both the cancer cell lines, as a preventive treatment, but also as a therapeutic treatment when the mice were injected with C1-OVA LNPs after cancer cell line inoculation.

Among the different intracellular pathways, the STING pathway (stimulator of interferon genes) started to show potential advantageous effects for DCs maturation and antigen presentation. STING is an adaptor protein involved in the cytosolic surveillance system [103,104], which was described in association with cGAS (cyclic GMP-AMP synthase) to sense exogenous DNA and induce a strong type I interferon (IFN) innate immune response [105]. Moreover, it has been shown that the activation of STING leads to the production of IL-12 (involved in induction and maintenance of a Th1-biased immune response [106], but also in the expression of CD40 and CD86, which are considered as activation markers in DCs [107].

Miao et al. screened a library of different lipids used for LNPs and found that heterocyclic lipids can lead to an activation of the STING pathway [22]. In this study, they used a one-step three-component reaction to produce up to 1000 lipid formulations and decipher the best lipids for mRNA delivery in combination with strong immune activation. They first treated HeLa cells with LNPs encapsulating Fluc (firefly luciferase) and observed enhancement in the delivery and protein expression of the lipids with longer alkyls chains and reduced saturation. Then, they treated primary APC with LNPs encapsulating mLuc and identified two candidates with a specific ketone and an ester group, which transfect APCs and HeLa cells similarly. Those two candidates were confirmed to show the best results in vivo for both subcutaneous and intramuscular injections. The structure similarity between the two candidates are as follows: two amines in the polar head group spaced three carbons apart; no hydroxyl group; the presence of a tertiary amine. They then treated an ovalbumin (OVA)-expressing B16F10 mouse melanoma model with LNPs encapsulating OVA mRNA (mOVA), to test the adaptive immune response and anti-tumor efficacy of the two lipids candidates. They show that lipids with a heterocyclic amine head group induced a significantly higher antigen-specific cytotoxic T lymphocyte response, with robust IFN-γ secretion, in comparison with the linear head group. Further optimization of the lipid structure shows that even empty cyclic LNPs were also able to upregulate the dendritic cell activation marker (CD40, MHCII), up to two–three-fold in comparison to linear LNPs. When using cyclic LNPs (mRNA loaded or empty) on STING, for IFN receptor (IFNa/b-R) KO BMDCs, no maturation was observed in opposition to wild-type cells, but this was not the case for linear LNPs, both in KO and wild-type conditions, indicating that adjuvant effect is dependent on cyclic lipids and mediated by the STING pathway.

### 2.3. Other Mechanism to Take in Consideration Regarding LNPs Distribution

In addition to delivery system-specific transfection efficiency, the same system does not transfect all tissues and cell lines equally. Although differences in tissue transfection can be attributed to different biodistribution [3], differences in mRNA transfection amongst cell types or cell lines have less been studied.

Using tetrasulfide-incorporated dendritic mesoporous organosilica nanoparticles, modified with PEI (DMONs-PEI), Wang et al. evidenced a correlation between intracellular glutathione (GSH) levels and suppression of mRNA translation. They compared EGFP mRNA delivery in hard-to-transfect RAW246.7 murine macrophages and easily transfected human embryonic kidney HEK293 cells, using either DMON-PEI or PEI-organosilica nanoparticles without disulfide bonds (DMSN-PEI). They applied this knowledge to artificially decrease GSH levels for mRNA delivery; the transfection of RAW246.7 cells with DMON-PEI induced a 35% decrease in GSH levels, which enhanced the percentage of EGFP-positive cells from 22% using DMSN-PEI to 71% using DMON-PEI. Interestingly, the enhanced mRNA translation was not due to increased cellular uptake, but was attributed to GSH depletion mediated by the activation of ribosomal protein S6, which is a critical regulator of ribosome biogenesis.

Cheng et al. showed a selective organ-targeting (SORT) property of LNPs without any use of ligand, but based on different formulations [108]. They produced different LNPs to deliver mRNA into the spleen, the lung or the liver (with applications such as CRISPR/Cas9 editing system). This selective organ expression of mRNA, still yet misunderstood, shows, for example, that a home-made mRNA–LNPs formulation without DOTAP could promote luciferase activity exclusively in the liver after I.V administration. An increasing percentage of DOTAP, up to 10–15%, shifted the expression to the spleen, and to the lung when 50% of DOTAP was incorporated.

Islam and colleagues delivered mRNA encoding the tumor suppressor PTEN (phosphatase and tensin homologue deleted on chromosome 10) using a polymer hybrid lipid nanoparticle for prostate cancer therapy [109]. Self-assembly of mRNA with G0-C14 cationic lipid and PLGA (poly (lactic-coglycolic acid)) led to the formation of a polymer–lipid hybrid core, which was coated with a lipid PEG shell coating, yielding neutral NP (6 mV) with 120 nm. These NP were able to transfect 80% PC3 prostate cancer cells via macropinocytosis. Because they wanted to deliver PTEN mRNA by intravenous injection to both primary tumors and metastasis, the authors compared the following two PEG lipid shells: either ceramide-PEG, which are rapidly detached from the surface of NP, or more stable DSPE-PEG (1,2-distearoyl-sn-glycero-3-phosphoethanolamine-N-amino-PEG) [110]. The PEG chemistry had a dramatic impact on mRNA blood residence after intravenous injection. Upon 30 min post-injection, only 1% naked mRNA remained in the mice blood, 4% mRNA for ceramide-PEG-coated NP and 30% mRNA for DSPE-PEG-coated NP. This striking difference in blood circulation yielded more mRNA accumulation in prostate cancer xenograft tumors for DSPE-PEG-coated NP, resulting in therapeutic activity in both xenograft and disseminated metastasis models. In another study, the authors have selected the most stable DSPE-PEG in LNPs composition for tumor accumulation, based on the EPR (enhanced permeability and retention) effect. Those LNPs target hepatocytes via ApoE interaction in the bloodstream and it has been shown that they require a fast removal of the PEG layer [111].

Among the optimization of LNPs as a pharmaceutic, a lot is conducted on mRNA stability and lipids composition. Still, to get optimal results in vivo, the delivery route employed should be studied, which can play a major role depending on the target (for example, systemic versus specific organ delivery, rapid expression vs. long-term expression, etc.). Little is known on this aspect; Pardi et al. tried to decipher which administration route is the best to get protein expression in mice [112]. In this study, stable LNPs, incorporating modified firefly luciferase mRNA (HPLC purified and composed with 1-methylpseudouridine), were injected at different concentrations into mice by the following six routes: intravenous; intraperitoneal; subcutaneous; intramuscular; intradermal; and intratracheal. The kinetics of proteins expression were studied. Intravenous, intraperitoneal, intramuscular and intratracheal administration of 5 µg mRNA–LNPs resulted in major expression into the liver, corresponding to a systemic delivery of the particles. Expression in the liver occurred strongly in the first 24 h and ceased in two to three days after injection. For intravenous and intraperitoneal, a relatively small fraction of activity was measured in the site of injection, but in a longer period (up to seven days after injection). For intramuscular and intratracheal administration, a significant signal was measured in the muscle and lung, also for up to seven days after injection. Subcutaneous and intradermal routes only resulted in activity measured at the site of injection, lasting six days and ten days, respectively. Regarding the kinetics of expression, all the routes of administration show a peak 4 h after injection, followed by a decrease, with intramuscular and intradermal showing the longest duration of translation. Dose–response studies were also performed with an injection of LNPs, containing 0.1 µg, 1 µg or 5 µg of mRNA. Intravenous, subcutaneous, intratracheal and intramuscular injections resulted in a linear dose–response. Interestingly, intraperitoneal and intradermal deliveries did not show an increase in activity as a function of mRNA dose, suggesting saturation of uptake and translational activity at all three doses. However, increasing the doses resulted in an extended duration of activity. Depending on the route of administration, different patterns of expression can be measured for the same mRNA–LNPs formulation, leading to another important parameter to take in charge for the use of LNPs as a versatile therapeutic for many diseases.

With live-cell imaging on single-cell array (LISCA), Reiser et al. [113] tried to decipher, at the single-cell scale, if a correlation exists between mRNA–LNPs delivery timing and expression efficiency, using liver carcinoma cells (HuH7 cell line) as a cellular model. Time-lapse measurement of EGFP expression was performed to detect very early events of protein expression after mRNA transfection into a microfluidic device. The team used a translation-maturation model to study mRNA expression over time (Figure 7). This model takes into consideration the delivery of mRNA, translation into EGFP proteins, and the required maturation of EGFP to render it fluorescent. Nevertheless, it considers that all mRNA are delivered at the same time, which can represent a bias. The delivery process of mRNA occurs over a period of only a few hours, starting approximately 10 min after incubation, with the average distribution in 2–3 h. When looking at the influence of non-specific protein adsorption on EGFP expression, different behaviors are observed when lipoplexes or LNPs are used. The incubation of mRNA–lipoplexes (made with Lipofectamine 2000^®^), with different concentrations of fetal bovine serum (FBS), results in a decrease in transfection efficiency (from 92% of transfected cells at 0% FBS to 31% at 10% FBS), with increase in onset time delivery (1.6 h at 0% FBS and 2.9 h at 10% FBS), and a decrease in expression rate. Interestingly, when LNPs are used (made using ionizable lipid Dlin-MC3-DMA), the opposite behavior is observed. Regarding these results, an essential parameter to take into consideration for the intracellular fate of the mRNA particles is the protein corona formed after interaction with biological fluids (e.g., FBS) [114,115]. Such protein corona can impact the surface and drastically change mRNA delivery and expression rate. For lipoplexes, proteins contained into FBS can interact with the particle to form a protein corona, e.g., due to cationic lipids and electrostatic interactions.

The influence of the protein corona has been well described for LNPs-mediated liver delivery. Notably, apolipoprotein E (ApoE) is well described for the protein corona of LNPs [116]. This interest in ApoE is high, since in addition to LNPs specifically targeting the liver, a lot of LNP formulations exhibit a relatively high amount of liver accumulation in vivo. In the blood circulation and among different functions, ApoE is involved in the transportation of lipids to the liver for recycling. Recently, Sebastiani et al. [117] showed that the binding of ApoE induce lipid rearrangement of the shell and the core of LNPs, potentially leading to a different endosomal escape. To study this effect, they produced LNPs made of deuterated lipids coupled with a small-angle neutron scattering (SANS) analysis, to decipher the composition and the global structure of the LNPs, but also more precisely the distribution of the lipids around the particle. After incubation with ApoE, rearrangement between the lipid in the shell and the core occurs, which leads to a drop in mRNA encapsulation 1 day post-incubation. More interestingly, cholesterol enrichment in the shell is observed. Even if more studies are still needed, it appears that ApoE interaction with LNPs can explain the liver accumulation, thanks to its biological role. Moreover, the difference in transfection efficiency could be linked to a change in the lipid composition of the shell/core after this interaction.

## 3. Quick Look to Other mRNA Loaded Formulations

LNPs are part of the latest innovation in terms of a lipidic formulation to deliver mRNA, thanks to the use of microfluidic tools to produce them, but many studies involved other well-known formulations, such as liposome, micelles, or polymers. Liposomes are vesicles formed from lipids (synthetic or natural), which formed a lipid bilayer and an internal space capable of encapsulating molecules such as small drugs [118]. In 1987, Felgner and coworkers used a cationic liposome for gene therapy (with DNA) for the first time, made with DOTMA (N-[1-(2,3-dioleyloxy)propyl]-N,N,N-trimethylammonium chloride), and in 1989, the same team performed transfection with mRNA [119,120]. Complexes made of liposomes and nucleic acids are called lipoplexes; the nucleic acid is embed through the bilayer, thanks to electrostatic interactions [121], which can impact the surface aspect and stability of the liposome, and then the endocytic behavior [122,123].

An interesting delivery platform to consider is light-triggered lipid formulation. In fact, after injection of classic LNPs (locally or systemically), there is no option to physically trigger the endosomal escape. Light-triggered formulations, thanks to specific wavelengths suitable for therapeutic use, allow a precise spatiotemporal release of the cargo inside the cell, which can be needed in some diseases to specifically target organs, or simply to decrease systemic toxicity. Up to date, most of the formulations are liposomes [124], but can also be lipopolyplexes [125], and they show promising results in LNPs formulation [126]. To be triggered by light, different strategies exist based on a photosensitive molecule, e.g., verteporfin (VP), which, under a specific wavelength illumination, can generate reactive oxygen species (ROS) that destabilize the endolysosomal membrane [127,128], or with indocyanine green (ICG), which can absorb and convert light energy into heat to destabilize the membrane [129]. Moreover, lipids can also be light sensitive, e.g., allowing the surface charge of a liposome to change [130], which can serve to enhance systemic circulation when neutral, and cellular uptake/endosomal escape when charged. With a high interest and encouraging results in cancer therapy, thanks to precise drug release, light-triggered formulations have been shown to be capable of nucleic acid release, such as DNA or oligonucleotides, which can lead to a promising platform for mRNA delivery [127,129].

Nevertheless, lipoplexes mRNA therapy could still be interesting for therapeutic purposes. Recently, we showed that co-delivery with lipoplexes of non-structural protein-1 (NS1) mRNA with bone morphogenetic protein-2 (BMP-2) mRNA increases the expression of BMP-2 and osteogenic differentiation into murine pluripotent stem cells [131]. Influenza A virus (IAV) non-structural protein 1 is a multifunctional protein, helping virus replication and virulence, which is known to interact with several proteins (e.g., RIG-I) to inhibit the activation of transcription factors (IRF3/7, NF-kB, etc.) required for IFNs production, leading to anti-viral response and reduced translational activity. C2C12-BRE/LUC cells that stably express luciferase reporter gene under BMP-2 responsive elements (BRE-luciferase), were transfected with lipoplexes containing 1 µg of mRNA with a different ratio between BMP-2 and NS1. When cells are transfected with BMP-2 mRNA alone, luciferase activity decreases by 65% after 48h, and almost completely (91%) after 72 h. When transfected with 0.25 µg of NS1 mRNA and 0.75 µg of BMP2, luciferase activity was 3.5- and 4.7-fold higher at 48 h and 72 h, respectively, and the decrease of luciferase activity was lower after 48 h (54%) and 72 h (85%) (Figure 8) [132].

Lipopolyplexes (or lipid–polymer hybrid nanoparticles) are ternary complexes formed between a nucleic acid, a polycation, and liposomes. In 1996, Huang et al. showed that the combination of cationic polymer (poly(L-lysine), protamine) with cationic liposome resulted in smaller and more stable particles, with increased transfection efficiency in comparison to lipoplexes [133]. Since then, several studies were performed to deliver different nucleic acids with lipopolyplexes, for example, DNA [56], interfering RNAs [134,135], and mRNA [5,6,136]. Recently mRNA-loaded lipopolyplexes have emerged as a suitable therapeutic that can be used for vaccination [3].

## 4. Future Directions

Most of the mRNA vaccines under phase II/III clinical trials are mRNA LNPs injected intramuscularly [3,11,33]. Such local administration would be well suited to light-triggered enhancement of endosomal escape, by the incorporation of a responsive unit, which is a feature already demonstrated for chemotherapy and DNA delivery [137,138,139].

Aside from cellular experiments, robust and predictive in vitro experiments are required for the screening of mRNA delivery systems. Zhang and colleagues used fluorescently labeled mRNA and fluorescence correlation spectroscopy (FCS) to understand the different transfection efficiencies of two frequently used transfection agents, Lipofectamine messenger MAX (LFM) and 22 kDa jetPEI (PEI) [140]. By measuring the diffusion coefficient of free mRNA, and mRNA complexed with LFM or PEI in Hepes buffer (20 mM, pH 7.4), they measured the association degree required for complexation. FCS indicated a sharp decrease in mRNA mobility between the free mRNA (200 kHz) and mRNA complexes (16 kHz). This decrease in mobility was accompanied with an increased mRNA fluorescence intensity, due to the entrapment of several mRNAs in the same assembly. The absence of free mRNA was confirmed by gel electrophoresis. Next, still using FCS, the stability of the complexes in biological fluids (human serum or ascetic fluids) and their ability to dissociate in cell lysates was evaluated. Although LFM complexes were as stable as PEI complexes after 1 h incubation at 37 °C in biological fluids, a striking difference was observed in cell lysates. Although 20–40% of mRNA LFM complexes were dissociated after 1 h at 37 °C, less than 5% of PEI mRNA complexes were dissociated. Finally, cellular experiments were performed to study transfection efficiency using SKOV-3 human ovarian cancer cells. Using cyanine 5-labeled mRNA encoding EGFP, both intracellular uptake and mRNA expression were measured. It was established that (i) close to 100% cells were positive for cyanine 5, 4 h after transfection with PEI or LFM; (ii) higher cyanine 5 fluorescence was detected with LFM over PEI (7000 MFI vs. 3000 MFI), suggesting more mRNA delivery per cell with LFM; and (iii) flow fluorescence correlation spectroscopy analysis of cyanine 5 mRNA signal in the cytoplasm revealed a 10-fold higher signal of mRNA–LFM complexes over PEI complexes, confirming the flow cytometry results. Such a dramatic difference in cytoplasmic mRNA delivery resulted in more EGFP-expressing cells, with LFM over PEI (93% vs. 60%), and a 25-fold higher EGFP expression of LFM complexes (5000 MFI vs. 200 for PEI). The correlation between dissociation ability in cell lysates and transfection efficiency strongly supports FCS analysis as a preliminary readout of mRNA delivery systems.

The mRNA field rapidly benefited from the development of therapies tailored to small RNAs, such as antisense RNA (e.g., Exondys 51^®^(eteplirsen), approved by the FDA in 2016 for the treatment of Duchenne muscular dystrophy) or small interfering RNA (e.g., Onpattro^®^ (patisiran), approved by the FDA in 2018 for the treatment of hereditary transthyretin amyloidosis) [141]. The following two mRNA vaccines including chemically modified nucleotides were approved in 2020 against SARS-Cov2: Comirnaty^®^(BNT162b2) from Pfizer-Biontech, and mRNA-1273 from Moderna Therapeutics [23]. These vaccines are both administered intramuscularly, but differ in lipids composition and mRNA dosage, as follows: 100 µg for the mRNA-1273, and 30 µg for Comirnaty [11,23]. The deployment of mRNA vaccines faces another challenge, as follows: mRNA–LNPs vaccines storage. The Comirnaty vaccine can be stored at −70 °C for 6 months and only 5 days between 2 °C and 8 °C, whereas mRNA-1273 can be stored at −20 °C for 6 months and at 2 °C to 8 °C for 30 days [11]. To date, it is presumed that the instability could be related to the lipids composition, but the interaction between lipids and mRNA could have a great impact as well.

These recent developments indicate that the immunogenicity and stability of therapeutic mRNA can be controlled by sequence optimization and the incorporation of modified nucleotides, resulting in efficient translation after local delivery. The remaining bottlenecks to the broad application of mRNA therapeutics appear to be the storage of formulated mRNA, targeted intracellular delivery, and intracellular processing. Over the past decade, mRNA has been implemented in the pipelines of Big Pharma (Pfizer, New York, NY, USA; GSK, Brentford, UK; Sanofi Pasteur, Lyon, France; Genentech, South San Francisco, CA, USA; Merck, Kenilworth, NJ, USA) and birthed healthcare technology giants (Moderna, Cambridge, MA, USA; Curevac, Tübingen, Germany; Biontech, Mainz, Germany). Undoubtedly, such sustained investments, along with a continuous flow of clinical trials in different therapeutic fields, together with research on mRNA technology all over the world, will turn mRNA into an affordable effective treatment modality.

## Figures and Tables

**Figure 1 pharmaceutics-13-00945-f001:**
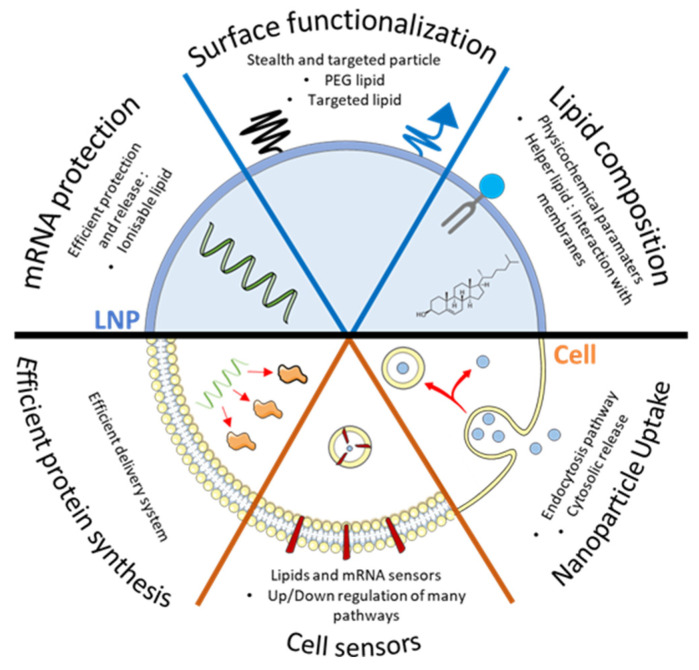
Critical parameter to consider regarding LNPs formulation and cellular interactions. To perform efficient transfection, LNPs should protect mRNA with ionizable lipid until its final destination and should be able to release it efficiently. Surface functionalization through incorporation of targeted lipid will help to reach specific organ/cells while PEGylated lipid will help the circulation of particles in vivo. Finally, incorporation of helper lipids will help either the formation of LNPs or its interaction with biological membrane. Altogether, those lipids need to trigger an efficient cellular uptake of the particle, which leads to the cytosol and avoids the lysosome, without triggering deleterious cell sensors, so that mRNA can lead to an efficient protein production in the cell.

**Figure 2 pharmaceutics-13-00945-f002:**
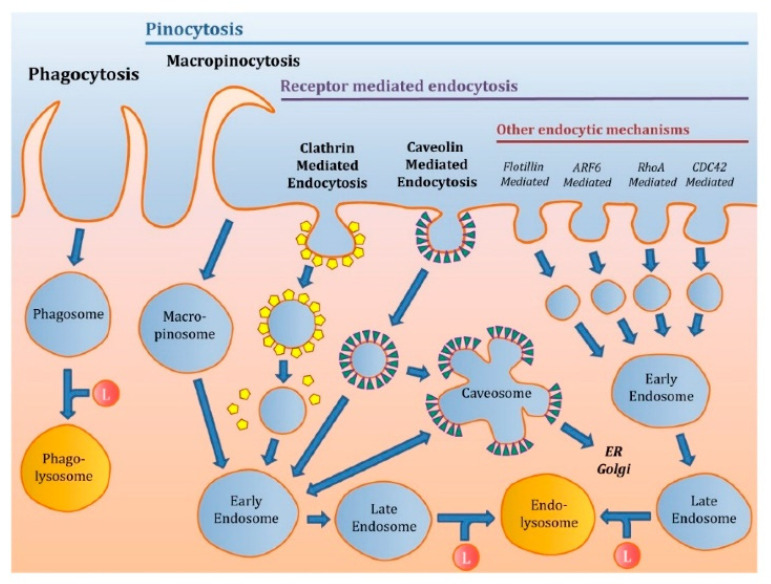
Principal modes of endocytosis, reproduced from [39], MDPI, 2020, under creative common license.

**Figure 3 pharmaceutics-13-00945-f003:**
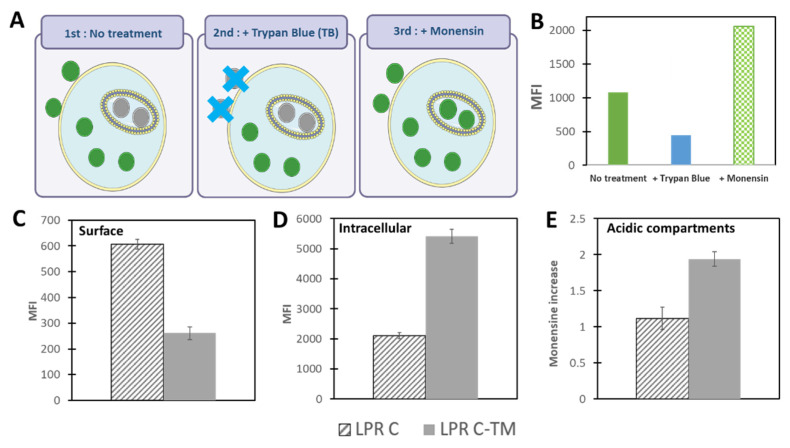
Flow cytometry study of mRNA NP uptake and localization. (**A**) Schematic representation of uptake study process. Cell-associated fluorescence intensity (MFI: mean fluorescence intensity) was measured in PBS (total florescence) after TB treatment (intra-cellular fluorescence) and after monensin treatment (fluorescence recovery in acidic compartments). (**B**) Histograms showing the effects of different treatments on recorded MFI. (**C**–**E**) corresponding to the cell surface-associated fluorescence, the intracellular fluorescence and the recovery of quenched fluorescence expressed and monensin increase. Shown are unpublished results corresponding to murine dendritic (DC2.4) cells incubated for 4 h with cationic mRNA–lipopolyplexes (LPR C) or cationic mRNA–lipopolyplexes bearing a trimannose ligand (LPR C-TM), without serum.

**Figure 4 pharmaceutics-13-00945-f004:**
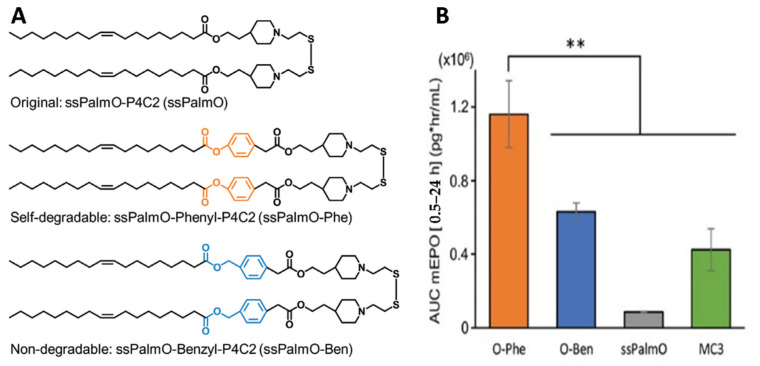
Improvement in mRNA expression in vivo by tuning the intracellular degradation. (**A**) Structures of the original lipid (ssPalmO), the lipid with a degradable phenyl ester (ssPalmO-Phe), and the lipid with a non-degradable benzyl ester (ssPalmO-Ben); (**B**) LNPs prepared with EPO mRNA were intravenously injected in mice (0.05 mg kg^−1^) before determination of blood EPO concentration by ELISA 24 h after injection (** *p* < 0.01). Reproduced from [73], Wiley, 2020, under creative common license.

**Figure 5 pharmaceutics-13-00945-f005:**
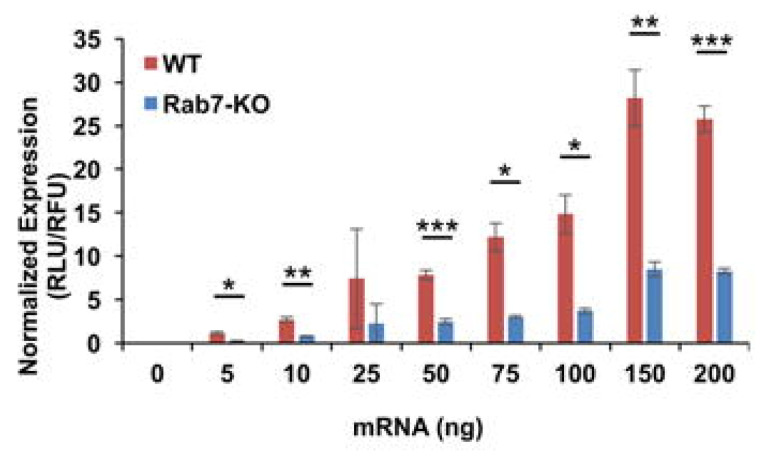
Late endosomes are essential for mRNA transfection wild-type (WT) and HAP1 cells with deleted Rab7 (Rab7-KO) were transfected with luciferase mRNA lipoplexes. A clear decrease in luciferase expression in Rab7-KO cells was evidenced. Adapted with permission from [88]. Copyright (2021) American Chemical Society. (* 0.05 ≥ *p* > 0.01, ** 0.01 ≥ *p* > 0.005, *** 0.005 ≥ *p*).

**Figure 6 pharmaceutics-13-00945-f006:**
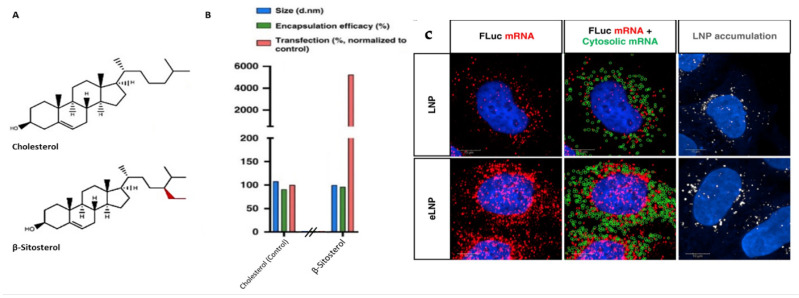
β-sitosterol enhanced mRNA-based gene transfection. (**A**) The structure of cholesterol and β-sitosterol. (**B**) Particles made of cholesterol or β-sitosterol were screened for size (nm), mRNA encapsulation (percent) and transfection efficiency (200 ng of mRNA). (**C**) Endosomal escape was visualized using smFISH. Representative fluorescent images showing mRNA, LNPs, and image analysis after delivery with LNPs (control) or eLNP (containing C-24 alkyl phytosterols) in HeLa cells. Adapted from Patel et al. [77], Nature Portfolio, 2020, under creative common license.

**Figure 7 pharmaceutics-13-00945-f007:**
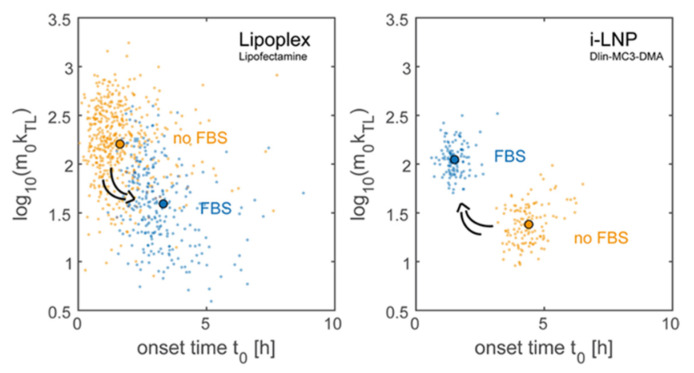
Parameter estimation of single-cell expression time courses using the translation-maturation model. Protein adsorption on the nanocarrier’s surface has the opposite effect on ionizable lipid nanoparticles i-LNPs compared to lipoplexes. Scatterplots of expression onset time t0 vs. expression rate m0kTL of lipoplexes (left) and i-LNPs (right) with (orange) and without serum (blue). The arrows indicate the opposite effect on transfection efficiency and timing induced by addition of FBS. Each data point corresponds to a single cell. The median value is indicated as full dot. Adapted from Reiser et al., [113], Oxford University Press, 2019, under creative common license.

**Figure 8 pharmaceutics-13-00945-f008:**
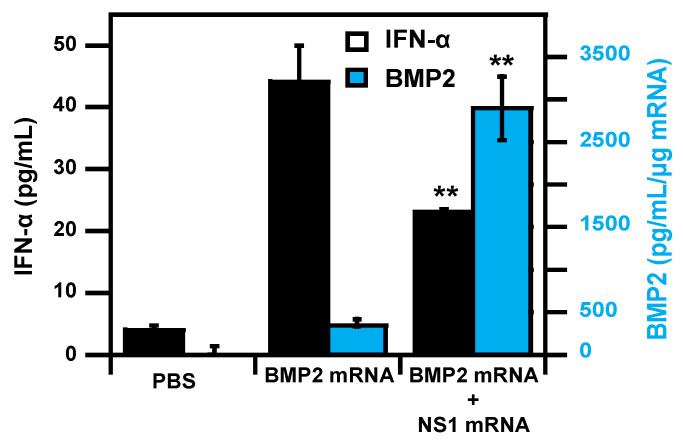
Co-transfection of osteoprogenitor cells with BMP2 mRNA and NS1 mRNA resulted in decreasing type I interferon response together with enhanced therapeutic BMP2 mRNA expression. Adapted from [132], Elsevier, 2021, with permission. ** represents *p* < 0.01.

**Table 1 pharmaceutics-13-00945-t001:** Summarizing table of the different lipidic formulation reported in this review.

Main Subject of Study	Enhancement Strategy	Model	Results	Ref
Nanoparticle uptake	Anti-PV1 conjugation for endothelial cell targeting in combination with 160 nm LNPs.	In cellulo In vivo	Increasing lung accumulation in combination with a 50-fold increase in mRNA expression compared to standard LNPs.	[50]
Nanoparticle uptake	Incorporation of tri-mannose lipid for mannose receptor targeting on dendritic cell, with lipopolyplexes.	In cellulo	Increase in the binding on the surface of dendritic cells was observed when liposome incorporate a tri-mannose moiety.	[60]
Endosomal escape enhancement	LNPs comprising lipid-like material ssPalmO for mRNA complexation and release after cleavage of disulfide bond by intracellular glutathione (GSH).	In vivo	LNPs made of ssPalmO-Phe (self-degradable ionizable phenyl ester lipid) showed the highest EPO blood level 24 h after transfection with EPO mRNA than LNPs made of non-degradable benzyl ester or with D-Lin-MC3-DMA.	[72,73]
Endosomal escape/transfection enhancement	Optimization of the length and saturation of the ionizable lipid tail.	In cellulo In vivo	Ionizable lipid with a length of 10 carbons exhibited highest luciferase expression in mice liver and spleen. Tails with more than 2 unsaturations showed lower luciferase expression.	[26,74]
Endosomal escape/transfection enhancement	Optimization of the polar head of the ionizable lipid, tertiary versus quaternary amine group.	In vivo	LNPs made with tertiary amine group lipids exhibited transfection on retinal pigment epithelial cells. LNPs based on quaternary amine group lacks transfection.	[75]
Endosomal escape/transfection enhancement	Screening of polar head and lipid tail for T lymphocyte transfection.	In vivo	Imidazole head and heteroatoms (O, S, Se) in tail exhibited highest transfection of primary T cells.	[76]
Endosomal escape/transfection enhancement	Screening of amino lipid series of ionizable lipids.	In cellulo In vivo	Lipid-5 exhibited a 6-fold increase in endosomal escape versus classic MC3-LNPs in cells. In vivo study showed a 5-fold enhancement of plasma level EPO in cynomolgus monkey after mRNA transfection versus MC3-LNPs.	[71]
mRNA translation enhancement	Screening of cholesterol analog.	In cellulo In vivo	LNPs made with cholesterol analog β-sitosterol exhibit equivalent uptake and endosomal escape, but a 48-fold enhancement in transfection versus cholesterol LNPs.	[77]
Immune sensors elusion	Combination of unmodified mRNA LNPs with drugs acting as inhibitors, as follows: ISRIB (eIF2a phosphorylation) and DXM (steroidal anti-inflammatory drug).	In-cellulo In vivo	Increase in mRNA translation was observed with both drugs in vitro, with different patterns. Co-delivery of DXM-palmitate and mRNA improve luciferase expression in mice liver.	[78]
Adjuvant effect of LNPs composition	Delivery of mRNA to APCs with LNPs made of lipid-like material allowing Toll-like receptor 4 activation for T-cell activation.	In cellulo In vivo	Lipid-like material induced maturation and activation of dendritic cell. Inhibitor of TLRs showed an effect specific to TLR4 signaling.	[79]
Adjuvant effect of LNPs composition	Screening of heterocyclic lipid for activation of STING pathway among APCs transfection.	In cellulo In vivo	Lipids with heterocyclic amine head group induced highest antigen-specific cytotoxic T lymphocyte response among OVA mRNA transfection. Empty LNPs made of cyclic group were also capable of DC activation, up to 2–3 fold versus linear LNPs, dependent on STING pathway.	[22]

Abbreviations: EPO: erythropoietin; ISRIB: integrated stress response inhibitor; DXM: dexamethasone; APCs: antigen-presenting cells; STING: stimulator of interferon genes.

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
