# Peer review of "Intracellular Routing and Recognition of Lipid-Based mRNA Nanoparticles"

_pharmaceutics, 2021, doi:10.3390/pharmaceutics13070945_

Round 1

Reviewer 1 Report

Thank you for inviting me to evaluate this review article. This review article addressed an important issue in lipid-based mRNA delivery field, intracellular trafficking of the lipid-nanoparticles (LNPs). The authors argue that three important behaviors (protein synthesis, cell sensors and uptake) of LNPs lies in LNP formulations, such as lipid composition. In each section, these three behaviors were respectively explained in detail with proper references, and most of them were well written and useful contribution for the mRNA delivery field. On the other hand, some paragraphs and sections were flawed as follows, and reduced the value of this interesting review. Thus, in my opinion, it requires major revision before ready for publication. I hope these comments will be helpful.

  1. There was no description about LNP in Abstract. Authors should mention it briefly.

  1. The Introduction was comprehensive, but I feel it is too general for the title of "lipid-based mRNA nanoparticle." The authors should add statements about the advantages or benefits of LNP for mRNA delivery in Introduction.

  1. Abbreviations, such as “lipid 5” in line 354, is unkind and makes it difficult to follow the explanation. It would be clearer to give each abbreviation used with its explanation.

  1. In line 221, authors referred the publications about the relationship between chemical structure of lipid and blood circulation profile of LNP. I think this paragraph is not related to the subject of this review article, or at least to the “intracellular trafficking of mRNA” section. I strongly recommend the authors either to omit this paragraph or to reconstruct it.

  1. In the paragraph from line 361, authors explained that in vivo efficiency was dictated by the influence of carbon length on pKa and surface ionization ability (reference 23). However, in my understanding, there was no correlation between pKa and in vivo efficiency in the reference 23. Please check the reference again, and correct the manuscript if necessary.

  1. Endosomal escape and cell sensors sections look like lists of experimental results. Due to the insufficient connection among each paragraph, the section lacks clarity and sharpness, and is poorly organized. Authors should reorganize these sections and provide connection and/or critique to give anything useful to readers.

  1. The paragraph from 623 explained mesoporous silica nanoparticles, not lipid-based nanoparticles. I do not understand why authors included this paragraph and what meaning of it.

Author Response

Reviewer 1 :

Comments and Suggestions for Authors

Thank you for inviting me to evaluate this review article. This review article addressed an important issue in lipid-based mRNA delivery field, intracellular trafficking of the lipid-nanoparticles (LNPs). The authors argue that three important behaviors (protein synthesis, cell sensors and uptake) of LNPs lies in LNP formulations, such as lipid composition. In each section, these three behaviors were respectively explained in detail with proper references, and most of them were well written and useful contribution for the mRNA delivery field. On the other hand, some paragraphs and sections were flawed as follows, and reduced the value of this interesting review. Thus, in my opinion, it requires major revision before ready for publication. I hope these comments will be helpful.

  1. There was no description about LNP in Abstract. Authors should mention it briefly.

We agree with the Reviewer, we added the sentence “We will focus on lipid nanoparticles (LNP) as they are the most advanced nanocarriers for the delivery of mRNA.” In the Abstract.

  1. The Introduction was comprehensive, but I feel it is too general for the title of "lipid-based mRNA nanoparticle." The authors should add statements about the advantages or benefits of LNP for mRNA delivery in Introduction.

We do agree with the Reviewer, the following paragraph has been added in the Introduction:

“The most advanced lipid-based NP for mRNA delivery are lipid nanoparticles (LNP) (re-viewed in [11, 15, 23]). LNPs are prepared by microfluidic mixing of lipids in ethanol and mRNA in acidic buffer (pH ≤ 4.0). Lipids include an ionizable lipid (pKa < 7) that will be protonated at acidic pH to condense mRNA and release mRNA inside the cells, cholester-ol for stabilization, an helper lipid for endosomal escape (DOPE (1,2-dioleoyl-sn-glycero-3-phosphatidyl-ethanolamine) or DSPC (distearoylphosphati-dylcholine) and a PEGylated (polyethylenglycol) lipid to prevent aggregation of LNPs.[3] LNPs combine several advantages: high mRNA encapsulation efficiency, can be injected by various routes, preclinical and clinical proof of activity and, stability under storage (re-viewed in [3, 11, 24]). Morevover, their production is reproducible and several techniques exist for large scale production of mRNA LNPs.[25] Accordingly, our review will focus on LNP-mediated delivery of mRNA.”

The following references have been added:

Buschmann, M.D., et al., Nanomaterial delivery systems for mRNA vaccines. Vaccines, 2021. 9(1): p. 65.

Evers, M.J., et al., State‐of‐the‐Art Design and Rapid‐Mixing Production Techniques of Lipid Nanoparticles for Nucleic Acid Delivery. Small Methods, 2018. 2(9): p. 1700375.

  1. Abbreviations, such as “lipid 5” in line 354, is unkind and makes it difficult to follow the explanation. It would be clearer to give each abbreviation used with its explanation.

We thank the Reviewer for this comment, however, “lipid 5” is not an abbreviation but the name of the lipid. Please see below a Table of ionizable lipids names and structures from Buschmann et al., Nanomaterial Delivery Systems for mRNA Vaccines. Vaccines 2021, 9, 65.https://doi.org/10.3390/vaccines 9010065

 All abbreviations were expanded upon first notice.

  1. In line 221, authors referred the publications about the relationship between chemical structure of lipid and blood circulation profile of LNP. I think this paragraph is not related to the subject of this review article, or at least to the “intracellular trafficking of mRNA” section. I strongly recommend the authors either to omit this paragraph or to reconstruct it.

We thank the Reviewer for carefully reading our manuscript and the work cited. We have taken the decision to move this paragraph to section 2.3 “Other mechanism to take in consideration regarding LNP distribution” as this part contains information such as the analysis of impact of different administration routes on transfection efficiency.

  1. In the paragraph from line 361, authors explained that in vivo efficiency was dictated by the influence of carbon length on pKa and surface ionization ability (reference 23). However, in my understanding, there was no correlation between pKa and in vivo efficiency in the reference 23. Please check the reference again, and correct the manuscript if necessary.

We thank the Reviewer for carefully reading our manuscript and the work cited. We have now re-read the reference 23 (now reference 26) and revised the sentence accordingly:

The sentence:

“In vivo efficacy was dictated by the influence of carbon length on pKa and surface ionization ability, not by LNP size or mRNA entrapment efficiency.”

Was edited to:

“In vivo efficacy was dictated by the influence of carbon length on amplitude of protonation, not by LNP size or mRNA entrapment efficiency.

  1. Endosomal escape and cell sensors sections look like lists of experimental results. Due to the insufficient connection among each paragraph, the section lacks clarity and sharpness, and is poorly organized. Authors should reorganize these sections and provide connection and/or critique to give anything useful to readers.

We thank the Reviewer for carefully reading our manuscript. Sections have been reorganized for a more fluent lecture, with addition of some connections. The added connections are in red in revised paper.

  1. The paragraph from 623 explained mesoporous silica nanoparticles, not lipid-based nanoparticles. I do not understand why authors included this paragraph and what meaning of it.

We do agree with the Reviewer and this paragraph has been removed.

Reviewer 1 :

Comments and Suggestions for Authors

Thank you for inviting me to evaluate this review article. This review article addressed an important issue in lipid-based mRNA delivery field, intracellular trafficking of the lipid-nanoparticles (LNPs). The authors argue that three important behaviors (protein synthesis, cell sensors and uptake) of LNPs lies in LNP formulations, such as lipid composition. In each section, these three behaviors were respectively explained in detail with proper references, and most of them were well written and useful contribution for the mRNA delivery field. On the other hand, some paragraphs and sections were flawed as follows, and reduced the value of this interesting review. Thus, in my opinion, it requires major revision before ready for publication. I hope these comments will be helpful.

  1. There was no description about LNP in Abstract. Authors should mention it briefly.

We agree with the Reviewer, we added the sentence “We will focus on lipid nanoparticles (LNP) as they are the most advanced nanocarriers for the delivery of mRNA.” In the Abstract.

  1. The Introduction was comprehensive, but I feel it is too general for the title of "lipid-based mRNA nanoparticle." The authors should add statements about the advantages or benefits of LNP for mRNA delivery in Introduction.

We do agree with the Reviewer, the following paragraph has been added in the Introduction:

“The most advanced lipid-based NP for mRNA delivery are lipid nanoparticles (LNP) (re-viewed in [11, 15, 23]). LNPs are prepared by microfluidic mixing of lipids in ethanol and mRNA in acidic buffer (pH ≤ 4.0). Lipids include an ionizable lipid (pKa < 7) that will be protonated at acidic pH to condense mRNA and release mRNA inside the cells, cholester-ol for stabilization, an helper lipid for endosomal escape (DOPE (1,2-dioleoyl-sn-glycero-3-phosphatidyl-ethanolamine) or DSPC (distearoylphosphati-dylcholine) and a PEGylated (polyethylenglycol) lipid to prevent aggregation of LNPs.[3] LNPs combine several advantages: high mRNA encapsulation efficiency, can be injected by various routes, preclinical and clinical proof of activity and, stability under storage (re-viewed in [3, 11, 24]). Morevover, their production is reproducible and several techniques exist for large scale production of mRNA LNPs.[25] Accordingly, our review will focus on LNP-mediated delivery of mRNA.”

The following references have been added:

Buschmann, M.D., et al., Nanomaterial delivery systems for mRNA vaccines. Vaccines, 2021. 9(1): p. 65.

Evers, M.J., et al., State‐of‐the‐Art Design and Rapid‐Mixing Production Techniques of Lipid Nanoparticles for Nucleic Acid Delivery. Small Methods, 2018. 2(9): p. 1700375.

  1. Abbreviations, such as “lipid 5” in line 354, is unkind and makes it difficult to follow the explanation. It would be clearer to give each abbreviation used with its explanation.

We thank the Reviewer for this comment, however, “lipid 5” is not an abbreviation but the name of the lipid. Please see below a Table of ionizable lipids names and structures from Buschmann et al., Nanomaterial Delivery Systems for mRNA Vaccines. Vaccines 2021, 9, 65.https://doi.org/10.3390/vaccines 9010065

 All abbreviations were expanded upon first notice.

  1. In line 221, authors referred the publications about the relationship between chemical structure of lipid and blood circulation profile of LNP. I think this paragraph is not related to the subject of this review article, or at least to the “intracellular trafficking of mRNA” section. I strongly recommend the authors either to omit this paragraph or to reconstruct it.

We thank the Reviewer for carefully reading our manuscript and the work cited. We have taken the decision to move this paragraph to section 2.3 “Other mechanism to take in consideration regarding LNP distribution” as this part contains information such as the analysis of impact of different administration routes on transfection efficiency.

  1. In the paragraph from line 361, authors explained that in vivo efficiency was dictated by the influence of carbon length on pKa and surface ionization ability (reference 23). However, in my understanding, there was no correlation between pKa and in vivo efficiency in the reference 23. Please check the reference again, and correct the manuscript if necessary.

We thank the Reviewer for carefully reading our manuscript and the work cited. We have now re-read the reference 23 (now reference 26) and revised the sentence accordingly:

The sentence:

“In vivo efficacy was dictated by the influence of carbon length on pKa and surface ionization ability, not by LNP size or mRNA entrapment efficiency.”

Was edited to:

“In vivo efficacy was dictated by the influence of carbon length on amplitude of protonation, not by LNP size or mRNA entrapment efficiency.

  1. Endosomal escape and cell sensors sections look like lists of experimental results. Due to the insufficient connection among each paragraph, the section lacks clarity and sharpness, and is poorly organized. Authors should reorganize these sections and provide connection and/or critique to give anything useful to readers.

We thank the Reviewer for carefully reading our manuscript. Sections have been reorganized for a more fluent lecture, with addition of some connections. The added connections are in red in revised paper.

  1. The paragraph from 623 explained mesoporous silica nanoparticles, not lipid-based nanoparticles. I do not understand why authors included this paragraph and what meaning of it.

We do agree with the Reviewer and this paragraph has been removed.

Reviewer 2 Report

The presented review on intracellular routing of mRNA as delivered by lipid-based NPs is well prepared, actuall and very interesting analysis of the very important topic. I like that authors first describe the problem, than give some important examples with sufficient amount of specific information, and finallu make some conclusions and future perspectives. In my opinion this review is very important to read for everybody from the area.

The number of recent citet literature (2019-2021) is more than 30 %.

Minor comments:

1) Page 8, line 284; Page 13, line 517; Page 15, line 618; Page 17, line 722; Page 18, line 768. There are no numbering for the sections headings

2) References: 25, 34, 74, 89, 109, 128 do not contain important information. Please check the references appearance.

3) Possibly it would be nice to add some words and refs about perspectivness and intracellular fate of light-triggered liposomes for mRNA delivery.

Author Response

Please see the attachment in the box

Reviewer 3 Report

Among other non-viral gene delivery systems, lipid-based vectors provide a promising alternative to viral vectors. The review summarizes the studies in the field, providing a detailed overview of the intracellular routing of mRNA lipoplexes and strategies utilized to overcome multiple intracellular barriers for mRNA delivery. However, the text needs further sharpening and cannot be accepted for publication in its present form.

MAJOR

  1. Although the review is devoted to strategies for improving intracellular routing of mRNA-based lipoplexes, it should be noted that the whole idea to describe overcoming intracellular barriers of mRNA transport is one-sided unless a description of extracellular barriers will be included in the review. It is difficult to improve in vivo mRNA delivery without approaches to cross extracellular barriers. Figure 1 shows the critical parameter of LNPs cellular interactions, which are also important for crossing some of the extracellular barriers; however, the description in the text is poor.
  2. Summarizing table is needed for studied carriers and approaches.

MINOR

  1. Misprints should be corrected and language checked. E.g. line 128 fuse fuse; 131 rho-GTP-ases; 269 degrada-tion etc.
  2. Not a particularly important matter, but I just wondered about the authors' decision to use the term in cellulo versus in vitro? Some people like to use in cellulo to differentiate work done in cell cultures, say, from work using cell extracts or purified protein, whereas others insist both should be referred to as in vitro experiments.
  3. In some subchapters numbering is absent.

Author Response

Plase see the attachment.

Round 2

Reviewer 3 Report

I am glad to inform the authors that all the answers are given and satisfactory. Also, I do not recommend changing the term in cellulo to in vitro.